# Enhancing the First-Pass Effect in Acute Stroke: The Impact of Stent Retriever Characteristics

**DOI:** 10.3390/jcm13113123

**Published:** 2024-05-26

**Authors:** Eduardo Murias, Josep Puig, Carmen Serna-Candel, Eva María Gonzalez, Manuel Moreu, Elvira Jiménez-Gómez, Luis SanRoman, Fernando Aparici-Robles, Mikel Terceño, Antonio Jesús Mosqueira Martínez, Sonia Aixut, Veredas Romero, Jose Carlos Mendez, Antonio Sagredo-Barra, Yeray Aguilar, Mariano Espinosa de Rueda, Miguel Angel Castaño Blázquez, Saima Bashir, José Rodríguez Castro, Alfonso Lopez-Frías, Jose María Jiménez, Juan Chaviano, Victor Maestro, Javier Manso, Antonio Lopez-Rueda, Sebastià Remollo, Lluis Morales-Caba, Marc Comas-Cufí, Pedro Vega

**Affiliations:** 1Facultad de Medicina y Ciencias de la Salud, Universidad de Oviedo, 33007 Oviedo, Spain; muriaseduardo@uniovi.es; 2Hospital Clínic de Barcelona and IDIBAPS, 08023 Barcelona, Spain; lroman1@clinic.cat; 3Hospital General Universitario de Alicante Doctor Balmis, 03010 Alicante, Spain; csernacandel@gmail.com (C.S.-C.); antonio.sagredo.b@gmail.com (A.S.-B.); 4Hospital Universitario De Cruces, 48903 Bilbao, Spain; evagonzalezdiaz@yahoo.com (E.M.G.); x_manso@hotmail.com (J.M.); 5Hospital Clínico San Carlos, 28040 Madrid, Spain; manumoreu@gmail.com (M.M.); alfonsolflj@gmail.com (A.L.-F.); 6Hospital Universitario Reina Sofía, 14004 Cordoba, Spain; elvirajimenezgomez@gmail.com (E.J.-G.); veredasromero@gmail.com (V.R.); 7Hospital Universitario y Politécnico La Fe, 46009 Valencia, Spain; aparici_fer@gva.es (F.A.-R.); lluismoralescaba@gmail.com (L.M.-C.); 8Hospital Universitario de Girona Doctor Josep Trueta, 17007 Girona, Spain; mikelterceno@hotmail.com (M.T.); saima.sbv@gmail.com (S.B.); 9Complejo Hospitalario Universitario de Santiago de Compostela, 15706 Galicia, Spain; drmosqueiramartinez@gmail.com; 10Hospital Universitario de Bellvitge, L’Hospitalet de Llobregat, 08907 Barcelona, Spain; sonia.aixut@gmail.com (S.A.); alrueda81@hotmail.com (A.L.-R.); 11Hospital Universitario Ramon y Cajal, 28034 Madrid, Spain; jmendezce@gmail.com; 12Complejo Hospitalario Universitario Insular Materno Infantil, 35016 Las Palmas de Gran Canaria, Spain; yeraka@gmail.com; 13Hospital Universitario Virgen de la Arrixaca, 30120 Murcia, Spain; mm.espinosa@gmail.com; 14Hospital Universitario de Salamanca, 37007 Castilla-Leon, Spain; macb1981@gmail.com; 15Hospital Universitario Central de Asturias, 33011 Asturias, Spain; jorocas03@gmail.com (J.R.C.); jjimenezp87@gmail.com (J.M.J.); juanchaviano.grajera@gmail.com (J.C.); v.carradas@gmail.com (V.M.); peveval@yahoo.es (P.V.); 16Hospital Universitario Germans Trias i Pujol, Badalona, 08916 Barcelona, Spain; sremollo@gmail.com; 17Departamento de Informática, Matemática Aplicada y Estadística, Universidad de Girona, 17003 Girona, Spain; marc.comas@udg.edu

**Keywords:** thrombectomy, stroke, endovascular, stentriever

## Abstract

**Introduction:** Although stentrievers (SRs) have been a mainstay of mechanical thrombectomy (MT), and current guidelines recommend the use of SRs in the treatment of large vessel occlusion stroke (LVO), there is a paucity of studies in the literature comparing SRs directly against each other in terms of mechanical and functional properties. Timely access to endovascular therapy and the ability to restore intracranial flow in a safe, efficient, and efficacious manner have been critical to the success of MT. This study aimed to investigate the impact of contemporary SR characteristics, including model, brand, size, and length, on the first-pass effect (FPE) in patients with acute ischemic stroke. **Methods:** Consecutive patients with M1 occlusion treated with a single SR+BGC were recruited from the ROSSETTI registry. The primary outcome was the FPE that was defined as modified (mFPE) or true (tFPE) for the achievement of modified thrombolysis in cerebral infarction (mTICI) grades 2b-3 or 3 after a single device pass, respectively. We compared patients who achieved mFPE with those who achieved tFPE according to SR characteristics. **Results:** We included 610 patients (52.3% female and 47.7% male, mean age 75.1 ± 13.62 years). mFPE was achieved in 357 patients (58.5%), whereas tFPE was achieved in 264 (43.3%). There was no significant association between SR characteristics and mFPE or tFPE. Specifically, the SR size did not show a statistically significant relationship with improvement in FPE. Similarly, the length of the SR did not yield significant differences in the mFPE and tFPE, even when the data were grouped. **Conclusions**: Our data indicate that contemporary SR-mediated thrombectomy characteristics, including model, brand, size, and length, do not significantly affect the FPE.

## 1. Introduction

Mechanical thrombectomy (MT) is the current standard of care for patients with acute ischemic stroke and large-vessel occlusion (LVO), aiming to increase functional outcomes with early reperfusion. Multiple randomized trials have demonstrated the benefits of endovascular thrombectomy for LVO strokes, with studies comparing the best medical treatment to endovascular treatment and confirming MT’s advantage in anterior circulation LVO cases [1,2,3,4,5,6]. The results were definitive, showing a significant increase in the proportion of patients who were alive and independent at three months.

MT is an endovascular procedure that involves recanalization of an intracranial occlusion by removing the thrombus using a retrievable stent, aspiration catheter, or a combination of both techniques. The procedure is typically performed via transfemoral arterial access but can also be performed through the radial artery. A biaxial or triaxial catheter system is used, with progressively smaller caliber catheters inserted within the distal vasculature. A large-bore guide catheter is positioned proximally in the target great vessel, and digital subtraction angiography is acquired after the injection of iodinated contrast to confirm large-vessel occlusion. Two main techniques are used for clot retrieval: direct aspiration and stent retrieval. The direct aspiration technique involves navigating an aspiration catheter to the occlusion and aspirating the clot [6], while the stent retrieval technique involves navigating a microwire and microcatheter beyond the occlusion, unsheathing the stent within the thrombus, and retrieving it along with the clot. The combined stent retrieval aspiration technique can be used by placing an aspiration catheter at the proximal margin of the thrombus [7].

Several studies suggest that stent retrievers (SRs) are the preferred treatment option for acute ischemic stroke with an LVO [8], with recent evidence showing the non-inferiority of stent retrieval compared to direct aspiration techniques. The COMPASS trial compared ADAPT and SR as first-use techniques and showed equal outcomes using either approach. While first-pass revascularization was achieved in 57% of ADAPT patients and 51% of SR patients, 85% of SR patients underwent suction aspiration in conjunction with SR [9]. Second-generation devices in the HERMES trials, such as Solitaire and Trevo SRs, demonstrated improved successful reperfusion rates of 71% compared to first-generation devices, as measured by mTICI grades 2b-3.

While initially the objective was to achieve successful reperfusion, measured by an mTICI score of 2b-3, complete reperfusion (TICI 3) was found to be associated with improved neurological outcomes, including greater functional improvement and reduced infarct growth at 90 days [10,11,12]. As a result, the focus of MT for anterior circulation has shifted to achieving TICI 2b-3 or, preferably, TICI 2c-3. The first-pass effect (FPE), which refers to achieving complete or near-complete reperfusion in a single pass, is a significant predictor of favorable outcomes and can reduce healthcare resource consumption and costs. The true FPE (tFPE) represents complete revascularization (mTICI 3) with a single pass of the device without rescue therapy [12], while the modified FPE (mFPE) refers to near-complete revascularization (mTICI 2b/3) without rescue therapy.

Stent retrievers (SRs) are metallic devices designed to remove clots from arterial occlusions at the site. Over time, SRs have become more sophisticated in design, leading to improved functionality. Currently, there is a diverse range of SRs available from various vendors with different shapes, sizes, and materials.

SRs are categorized based on their material composition, manufacturing technique, geometric configuration, and incorporated enhancements aimed at improving or facilitating their application (Table 1). Regarding size, the initial devices introduced to the market could be described as short, typically measuring up to 20 mm in length. Subsequently, their length increased to 40 mm and 60 mm. Consequently, the utilization of short devices is currently anecdotal, except in cases involving distal small-caliber vessels [13].

However, there are no specific guidelines for selecting SRs, and the choice depends on the discretion of the neurointerventionalist. Two studies found that longer SRs (30–40 mm) may offer better outcomes compared to shorter SRs (20 mm) in terms of complete clot removal and freedom from procedure-related complications [14,15]. Larger devices were also associated with a higher frequency of complete clot removal. Additionally, the use of longer SRs was found to be an independent predictor of better outcomes in internal carotid artery (ICA) and middle cerebral artery (MCA) occlusions. This suggests that longer retrievers provide a larger surface area for interaction with the clot, reducing the likelihood of leaving the clot behind. In previous studies, similar SRs have demonstrated FP rates of 34.8% and 40.5%. A longer stent retriever (4 × 40 mm) showed the highest frequency of FP compared to a larger diameter (6 × 30) and shorter stents (4 × 20 mm) in ICA, MCA-M1, and MCA-M2 occlusions [16,17].

A study by Yang et al. [18] showed no significant difference in the effectiveness of intra-arterial therapy based on the size of stent retrievers, except in patients with atherosclerosis, where better reperfusion was associated with the use of small-diameter SRs. In vitro experiments have shown that longer SRs achieve a higher FP in fibrin-rich clots. However, one study found no difference in reperfusion rates between 4 and 6 mm diameter SRs, while others found higher rates of modified FP with short SRs. [14,15,18,19,20,21,22]. There is currently no conclusive evidence to suggest that one SR model is better than others for FPE and final reperfusion grade.

In an effort to standardize endovascular treatment techniques and consider the utilization of the most current and advanced devices in this multicenter study, we aimed to compare tFPE and mFPE in isolated M1 occlusions treated with SRs and guided catheter-based thrombectomies (GCBs) of various sizes, lengths, and models. Our working hypothesis posits that larger and longer SRs yield superior outcomes compared to smaller and shorter ones.

## 2. Materials and Methods

The Rossetti registry is an ongoing investigator-initiated prospective study conducted across 15 Comprehensive Stroke Centers in Spain. The registry aims to gather de-identified demographic, clinical presentation, site-adjudicated angiographic, procedural, and outcome data from consecutive patients with acute ischemic stroke who have undergone MT. The primary objective of the Rossetti registry is to assess the effectiveness and safety of different MT techniques employed for anterior circulation LVO. The registry, which began in June 2019, incorporates the latest device technology available.

SRs were used exclusively to treat 1295 of the 3490 patients who were enrolled in the Rossetti registry through to April 2023. From this total, 610 patients who presented exclusively with the occlusion of the M1 segment of the middle cerebral artery (MCA), without extracranial tandem occlusion, and without a second thrombus in another artery, were included.

Age, sex, initial NIHSS score, side of occlusion, ASPECT score, and type of anesthesia were recruited. For the primary objective of this study, tFPE and mFPE were analyzed concerning the SR’s brand, model, diameter, and length. Subsequently, a grouping of variables was then performed relative to the first commercially available model to determine if its modification improved the objective variables of the study.

The devices analyzed included Solitaire X (Medtronic, Irvine, CA, USA), Catch + (Balt, Montmorency, France), Trevo XP and NXT ProVue (Stryker Neurovascular, Fremont, CA, USA), Preset (Phenox, Bochum, Germany), Neva (Vesalio LLC., Nashville, TN, USA) and Aperio (Acandis^®^, Pforzheim, Germany).

Angiographic revascularization was assessed using the modified thrombolysis in cerebral infarction (TICI) score in the final run of the angiography, measuring reperfusion in the downstream territory of the specific arterial occlusion, as follows: grade 0 = no reperfusion; grade 1 = antegrade reperfusion past the initial occlusion, but with limited distal branch filling and little or slow distal reperfusion; grade 2a = antegrade reperfusion of less than half of the occluded target artery previously ischemic of the downstream territory; grade 2b = antegrade reperfusion of more than half, but <90% complete antegrade reperfusion; grade 2c = near-complete reperfusion (90–99%) except for slow flow in a few distal cortical vessels or the presence of small distal cortical emboli; and grade 3 = complete antegrade reperfusion with the absence of visible occlusion in all distal branches [14,18,23,24,25,26].

The primary outcome of our study was to analyze the mFPE, defined as achieving near-complete revascularization of the large vessel occlusion and its downstream territory (mTICI 2b/3) with a single pass of the device, without the need for any additional rescue therapy. The second outcome is analyzing tFPE, defined as achieving complete revascularization of the LVO and its downstream territory (mTICI 3) with a single pass of the device, without the need for any additional rescue therapy [12,27]. Due to the nature of both measurements, the patients with tFPE are included in the mFPE.

Continuous variables are represented using the means, standard deviations, minimum and maximum values, and medians. Comparisons between continuous variables were performed using parametric tests. Categorical variables are described using relative and absolute frequencies. The chi-square test was used to assess the relationship between categorical variables. *p*-values below 0.05 were considered statistically significant. All statistical analyses were performed using R (version 3.6.1).

## 3. Results

We analyzed 610 patients [319 (52.3% female and 47.7 male) with a mean (SD) age of 75.1 ± 13.62 years. Left MCA occlusion was present in 52% of the patients, and 34.3% were treated under general anesthesia. mFPE was achieved in 58.5% and tFPE in 43.3% of patients. The characteristics of the study cohort are shown in Table 2.

Regarding the brand of SR used, the mFPE value ranged from 41.5% to 68%, whereas the tFPE ranged from 23% to 51% (*p* = 0.167). Table 3 presents the data for the brands. There were no significant differences according to the brand and model of SR for FPE. There were no statistically significant differences when comparing Solitaire X with the other SR brands (*p* = 0.971). In contrast, neither the diameter nor the length of the SR device was associated with better FPE rates (tFPE, *p* = 0.893; mFPE, *p* = 0.731 for diameter; tFPE, *p* = 0.815; mFPE, *p* = 0.306 for length), even when continuous variables or grouped in different measurement ranges were considered (Table 4 and Table 5). Finally, using the diameter and length as continuous variables, no significant relationship was found in either of the FPE categories (Pearson correlation *p* = 0.894 for device length, *p* = 0.600 for device diameter, and regression analysis of variance *p* = 0.687 for device length and *p* = 0.517 for device diameter).

## 4. Discussion

Most studies on SRs have been updated by vendors, so incorporating newer evidence into the literature is valuable, with a specific emphasis on achieving FPE. This outcome is associated with improved clinical outcomes, fewer procedural complications, reduced rates of hemorrhagic transformation, lower mortality rates, and reduced healthcare resource utilization and costs. Different models of SRs exhibit various morphologies, sizes, and lengths to improve FPE. Several published studies have analyzed the efficacy of SRs in relation to their length and size. Theoretically, longer and larger SRs are expected to enhance thrombus extraction by improving adhesion in occluded vessels and providing greater stability to the thrombus removal process. In vitro studies have demonstrated that longer SRs are associated with higher thrombus removal rates.

However, these results have been contradictory. Similar to our findings, some studies have shown that larger caliber and longer SRs do not significantly increase FPE rates compared to patients treated with thinner and shorter SRs. We were unable to demonstrate that the brand, model, size, or length of the SR are predictors of FPE, either with mFPE or tFPE. Even when comparing various brands with the Solitaire SR, considering the first SR on the market and its use in multiple clinical trials, we did not find any significant differences between them. This suggests that the modifications made by different manufacturers to the standard structure of the Solitaire SR may not have contributed to the higher FPE rate in our series. Further studies comparing the performance of 4 and 6 mm diameter Solitaire stent retrievers did not find differences in the outcomes of endovascular treatment [18,28].

In our study, the FPE rates we observed fell within the range reported by other groups (30–70%). It is worth noting that our study’s strengths include the collective experience of multiple institutions that encompass the Rossetti registry and the size of the sample we presented. Although we did not find any significant differences in FPE rates based on the type of SR used, other factors such as the properties and composition of the thrombus, technical aspects related to the position and manipulation of the stent, the anatomical and morphological factors of the arteries, and other factors may contribute to the FPE rate. In terms of thrombus composition, thrombectomy trials with simulated red (clumped red blood cells) and white (fibrin-based) clots of different sizes have been conducted. While all the tested devices were able to engage and displace small and medium white clots to some degree, they were unable to engage and displace large white clots. On the other hand, red clots were completely engaged but underwent significant fragmentation, which could lead to distal embolization. Future developments in thrombectomy devices will likely focus on improving FPE rates and following emerging trends and new frontiers in thrombectomy indications [19].

One consideration in our study that may result in different outcomes compared to previous studies is the characteristics and experience of the centers included. While some previous studies have suggested that the size and length of the SRs are crucial factors [29], we believe that the widespread adoption of endovascular treatment and increasing experience and safety of devices has led to the use of short stent retrievers (≤20 mm) becoming less common, comprising only about 30% of the sample, with the majority being devices >40 mm. As a result, the difference between short and medium devices is likely to be more significant than those between medium and long devices.

It is important to note that technical and other factors associated with the mechanical MT, which were not analyzed in our study, may affect the likelihood of FPE. Substudies using Rossetti registry data are currently investigating these issues. Moreover, it is difficult to conduct a precise comparison of thrombectomy performance due to the involvement of several operators with varying degrees of experience and technical expertise.

## 5. Conclusions

Our data indicate that the characteristics of SR-mediated thrombectomy, including model, brand, size, and length, do not affect FPE in patients with isolated M1 occlusion. It is possible that anatomical, technical, and thrombus-related factors may have a greater impact on FPE than SR-mediated thrombectomy.

## Figures and Tables

**Table 1 jcm-13-03123-t001:** A survey of current and historical SRs (adapted from 10).

SR	Manufacturer	Material	Fabrication Method	Configuration	Additions
First Generation					
Merci Retriever	Concentric Medical	Nitinol	Memory-shaped wire	5 helical spiral	N/A
Phenox	Phenox GmbH	Nitinol	Wire	N/A	Polyamide microfilaments
Catch	Balt Medical	Nitinol	Braided	Self-expanding basket	Radio-opaque markers
Second Generation					
Solitaire	Medtronic	Nitinol	Laser cut sheet	Closed cell, peak-peak	Radio-opaque platinum markers
Trevo	Stryker	Nitinol	Laser cut tube	Closed cell, peak-peak	Braided radio-opaque wires
pRESET	Phenox GmbH	Nitinol	Laser cut	Closed cell; distal helical slit	Radio-opaque markers
Aperio	Acandis	Nitinol	Laser Cut	Closed cell, peak-peak	Braided radio-opaque wires
Third Generation					
EmboTrap	Cerenovus	Nitinol	Two-layered laser cut	Inner: closed cell; Outer: open cell prox, closed cell distal	Radio-opaque markers
Tigertriever	Rapid Medical	Nitinol	Braided	Closed cell	Central wire with mechanical pulley for radial force
3D Revascularization Device	Penumbra	Nitinol	Laser cut tube	Closed cell inner chambers; open cell outer leaflets	Radio-opaque markers
Eric	Microvention	Nitinol	Laser cut	Closed cell, interconnected spherical cage design	N/A
Nimbus	Cerenovus	Nitinol	Laser cut	Proximal open cell spiral; Distal closed cell	Radio-opaque markers
NeVa	Vesalio	Nitinol	Laser cut	Hybrid-cell stent retriever	Three zones of working.

**Table 2 jcm-13-03123-t002:** Characteristics of the cohort.

Variable	Data
Gender (female), *n* (%)	319 (52.3%)
Age (years), mean ± SD	75.1 ± 13.62
NIHSS, mean ± SD	16.2 ± 6.2
M1-MCA occlusion (left), *n* (%)	316 (52.0%)
ASPECTS on baseline, median (IQR)	8 (3–10)
Sedation, *n* (%)	401 (65.7%)
General anesthesia, *n* (%)	209 (34.3%)
TICI 0 after first-pass, *n* (%)	169 (27.7%)
Modified first-pass effect (mFPE), *n* (%)	357 (58.5%)
True first-pass effect (tFPE), *n* (%)	264 (43.3%)

**Table 3 jcm-13-03123-t003:** Brand and model of the SR according to FPE.

**Brand**	***n* (%)**	**mTICI 0**	**mFPE**	**tFPE**
Solitaire, *n* (%)	290 (47.54%)	90 (30.1%)	131 (45.0%)	172 (41.5%)
Trevo, *n* (%)	133 (21.8%)	29 (21.5%)	74 (54.8%)	49 (36.3%)
Catch, *n* (%)	66 (10.8%)	18 (27.3%)	43 (65.2%)	34 (51.5%)
Embotrap, *n* (%)	64 (10.5%)	18 (28.1%)	34 (53.1%)	29 (45.3%)
Neva, *n* (%)	25 (4.1%)	5 (20%)	17 (68.0%)	8 (23.0%)
Others, *n* (%)	32 (5.3%)	9 (28.1%)	19 (59.3%)	13 (40.6%)
*p*-valor	-	0.212	0.457	0.167
**Model**	***n* (%)**	**mTICI 0**	**mFPE**	**tFPE**
Solitaire 6 × 40	101 (16.56%)	33 (32.7%)	58 (57.4%)	45 (44.6%)
Solitaire 4 × 20	96 (15.74%)	27 (28.2%)	63 (65.6%)	46 (47.9%)
Solitaire 4 × 40	78 (12.79%)	26 (33.4%)	42 (53.8%)	34 (43.6%)
Embotrap 5 × 37	50 (8.20%)	16 (32%)	24 (48%)	20 (40.0%)
Trevo 6 × 30 NXT	38 (6.23%)	10 (26.3%)	20 (52.6%)	19 (50.0%)
Trevo 4 × 30 XP	32 (5.25%)	7 (21.9%)	18 (56.3%)	12 (37.5%)
Trevo 4 × 35 NXT	25 (4.10%)	8 (33.3%)	11 (45.8%)	5 (20.0%)
Trevo 6 × 25 XP	24 (3.93%)	1 (4%)	18 (72%)	10 (41.7%)
Catch 5 × 35	23 (3.77%)	6 (26.1%)	16 (69.6%)	11 (47.8%)
Others	143 (23.4%)	-	-	-
p-valor	-	0.311	0.453	0.971

**Table 4 jcm-13-03123-t004:** Size and length of the SR according to FPE.

**Size [mm]**	***n* (%)**	**mFPE**	**tFPE**
3 y 4	304 (49.8%)	173 (56.9%)	144 (47.4%)
4.5 y 5	86 (14.1%)	55 (64.0%)	46 (53.5%)
6 y 6.5	220 (36.1%)	127 (57.7%)	108 (49.1%)
p-valor	-	0.731	0.893
**Length [mm]**	***n* (%)**	**mFPE**	**tFPE**
from 20 to 29	186 (30.5%)	113 (60.8%)	88 (47.3%)
from 30 to 39	201 (33.0%)	113 (56.2%)	102 (50.7%)
from 40 to 50	223 (36.%)	129 (57.8%)	108 (48.4%)
p-valor	-	0.306	0.815

**Table 5 jcm-13-03123-t005:** Devices used in this study. The first column shows the name of the device followed by its size; the first value corresponds to the diameter, and the second to the length. Devices named “others” are those with less than 1% of the sample that were grouped into a single group.

Device	Sample (*n* = 610)	(%)
Solitaire 6 × 40	101	16.56%
Solitaire 4 × 20	96	15.74%
Solitaire 4 × 40	78	12.79%
Embotrap 5 × 37	50	8.20%
Trevo 6 × 30 NXT	38	6.23%
Trevo 4 × 30 XP	32	5.25%
Trevo 6 × 25 XP	24	3.93%
Trevo 4 × 35 NXT	25	4.10%
Catch 5 × 35	22	3.61%
Catch 4 × 20	17	2.79%
Solitaire 6 × 24	15	2.46%
Catch 6 × 50	14	2.30%
Embotrap 65 × 45	13	2.13%
Trevo 4 × 20 XP	12	1.97%
Catch 6 × 40	9	1.48%
Neva 4 × 30	9	1.48%
Neva 45 × 29	8	1.31%
Other 4 × 30	5	0.82%
Other 6 × 40	5	0.82%
Catch 5 × 20	4	0.66%
Neva 4 × 22	4	0.66%
Catch 5 × 35	3	0.49%
Catch 6 × 30	3	0.49%
Other 4 × 20	3	0.49%
Other 4 × 35	3	0.49%
Aperio 45 × 30	2	0.33%
Aperio 45 × 40	2	0.33%
Neva 4 × 25	2	0.33%
Other 5 × 30	2	0.33%
Trevo 3 × 20	2	0.33%
Aperio 6 × 40	1	0.16%
Embotrap 5 × 22	1	0.16%
Neva 45 × 37	1	0.16%
Neva 45 × 44	1	0.16%
Other 4 × 40	1	0.16%
Other 5 × 33	1	0.16%
Other 6 × 30	1	0.16%

## Data Availability

The original contributions presented in the study are included in the article, further inquiries can be directed to the corresponding author.

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
