# Peer review of "Enhancing the First-Pass Effect in Acute Stroke: The Impact of Stent Retriever Characteristics"

_jcm, 2024, doi:10.3390/jcm13113123_

Round 1

Reviewer 1 Report

Comments and Suggestions for Authors The article "The Influence of Sound Retriever-Mediated Thrombectomy Characteristics on First-Pass Effect for Acute Stroke" explores the important topic of differences in efficiency between stent retrievers with different technological designs. With a large cohort, authors showed that the stent characteristics did not significantly impact the outcomes of mechanical thrombextraction. The article is well-written, with many details of the description, indicating the quality of the author's work, such as the hypothesis of the study. However, there are a few areas where improvements could be made: - It would be helpful to explicitly state whether mFPE and tFPE are nested categories, and if a patient may end up in both. - In the description of statistical methods, it would be beneficial to indicate whether a correction for multiple comparisons was used. - Finally, in the discussion section, explicitly mention the limitations of the study. - Please reread the text again. There are some unrelated sentences, for example line 176 - relatevely to what?

Author Response

"However, there are a few areas where improvements could be made: - It would be helpful to explicitly state whether mFPE and tFPE are nested categories, and if a patient may end up in both":  Due to the nature of both measurements, the patients with tFPE is included in the mFPE. We have included it in the text. 

"In the description of statistical methods, it would be beneficial to indicate whether a correction for multiple comparisons was used": We not use correction for multiple comparisions.

"Finally, in the discussion section, explicitly mention the limitations of the study":  It is important to note that technical and other factors associated with the mechanical MT, which were not analyzed in our study, may affect the likelihood of FPE. Substudies using Rossetti registry data are currently investigating these issues. Moreover, it is difficult to conduct a precise comparison of thrombectomy performance due to the involvement of several operators with varying degrees of experience and technical expertise.  We have added them to the text

"Please reread the text again. There are some unrelated sentences, for example line 176 - relatevely to what?" variables. We have included it in the text. 

Reviewer 2 Report

Comments and Suggestions for Authors

The manuscript titled “ Enhancing the First-Pass Effect in Acute Stroke: Impact of Con- temporary Stent Retriever Characteristics” led by Murias et al shows the different types of SR and their role in mechanical thrombectomy in M1 occlusion stroke patients. The current work is novel and provides an idea about SRs use in clinics for stroke patients. A few modifications are to be made in the current version of the manuscript.

1.     Title: Remove – from Con-temporary from the title

2.     Line 67: Reference 6 is incorrect.

3.     Carefully check the reference format. For example, line 56 cases. [1,5]., Line 71 thrombus. [7], and Line 78 [9]. Throughout the manuscript, references are represented incorrectly. Keep uniformity in references.

4.     Line 138-140: This sentence is inconclusive. Write whether patients are included or excluded.

5.     Line 174: male subjects’ details are missing.

6.     Line 40 and line 174 represent the 319 patients with male and female. Mention male or female and one gender sample size is missing. Therefore, the authors need to correct and add both gender details to the sample size.

7.     Line 176: mFPE was achieved in 58.5% and 43.3% of patients, respectively. Clearly state, what these % represent. Male or female?

8.     Authors details, such as state/province, and country are missing.

Comments on the Quality of English Language

Grammar and punctuations can be improved

Author Response

1.     Title: Remove – from Con-temporary from the title: Ok.

2.     Line 67: Reference 6 is incorrect. Ok

3.     Carefully check the reference format. For example, line 56 cases. [1,5]., Line 71 thrombus. [7], and Line 78 [9]. Throughout the manuscript, references are represented incorrectly. Keep uniformity in references. Ok.

4.     Line 138-140: This sentence is inconclusive. Write whether patients are included or excluded. "were included". OK

5.     Line 174: male subjects’ details are missing. (52.3% female and 47,7 male) OK

6.     Line 40 and line 174 represent the 319 patients with male and female. Mention male or female and one gender sample size is missing. Therefore, the authors need to correct and add both gender details to the sample size. (52.3% female and 47,7 male) OK

7.     Line 176: mFPE was achieved in 58.5% and 43.3% of patients, respectively. Clearly state, what these % represent. Male or female? mFPE was achieved in 58.5% and tFPE 43.3% of patients.

8.     Authors details, such as state/province, and country are missing. Ok.